# Caring for a Family Member with Psychosis or Bipolar Disorder Who Has Experienced Suicidal Behaviour: An Exploratory Qualitative Study of an Online Peer-Support Forum

**DOI:** 10.3390/ijerph192215192

**Published:** 2022-11-17

**Authors:** Paul Marshall, Steven Jones, Patricia Gooding, Heather Robinson, Fiona Lobban

**Affiliations:** 1Spectrum Centre for Mental Health Research, Division of Health Research, Faculty of Health and Medicine, Lancaster University, Lancaster LA1 4YT, UK; 2Division of Psychology and Mental Health, School of Health Sciences, Faculty of Biology, Medicine and Health, University of Manchester, Manchester M13 9PL, UK

**Keywords:** psychosis, bipolar disorder, suicidal behaviour, carers, families, qualitative

## Abstract

Background. The likelihood of suicidal behaviour is elevated amongst people with psychosis or bipolar disorder. This study aimed to understand how carers experience supporting family members with psychosis or bipolar disorder who have also experienced suicidal behaviour. Methods. A qualitative thematic analysis of online peer forum posts was carried out on the Relatives Education and Coping Toolkit (REACT) website, an online intervention for carers of people with psychosis and bipolar disorder. Analysis was based on 178 posts by 29 forum users. Posts were selected based on their relevance to suicidal behaviour. Results. Three themes were generated. “Suicide as the ultimate threat” highlights fears emerging from carers’ difficulties with understanding and managing suicidal behaviour. “Bouncing from one crisis to another” reflects carers’ experiences of recurring crises and the challenges of relying on emergency healthcare support. “It definitely needs to be easier to get help” emphasises carers’ desires to be acknowledged by healthcare professionals and included in support offered to service users. Conclusions. Digital platforms, including online forums, brief interventions such as safety planning, and interagency crisis models, hold the potential to meet carers’ needs in this context. However, further research is required to investigate the effectiveness and implementation of these approaches.

## 1. Introduction

Approximately 8.8 million people in the United Kingdom (UK) are involved in the provision of care for friends, family members, and other social contacts [1]. In the UK, healthcare policy explicitly recognises the value of this support by committing to develop the best practices for carer recognition and assistance as part of the National Health Service Long Term Plan [2]. Yet while community support for people experiencing mental health problems relieves considerable strain on national financial resources, it is often provided at significant personal cost to the carer [3,4]. Carers of people with psychosis and bipolar disorder consistently report high levels of distress, negative financial impacts, and limited access to social resources [5,6]. Qualitative studies with families of people with psychosis and bipolar disorder also highlight difficulties with accessing appropriate mental health services to support family members, especially during periods of crisis [7,8], and only 21% of those experiencing first episode psychosis in England receive family-based psychological interventions recommended by UK clinical guidance [9,10].

People with psychosis or bipolar disorder are significantly more likely than the general population to experience suicidal behaviour, including making attempts on their own life [11]. This is likely to be particularly challenging for carers [12]. Suicidal behaviour is associated with increased caregiver distress amongst people with schizophrenia [13], first episode psychosis [14], and bipolar disorder [15]. However, to date, mechanisms underpinning these associations remain unclear. In an interview study following suicide attempts by people experiencing psychosis, family members were largely unaware of any increased likelihood of suicidal behaviour prior to these suicide attempts, nor the need for additional mental health support [16]. This suggests that psychosis may present carers with additional barriers to recognising and responding to suicidal behaviour. Furthermore, interviews with service users and family members of people experiencing bipolar disorder investigating perceptions of healthcare support [17] highlight how carers’ desires to be actively involved in care offered to service users are not always met. Challenges included difficulties accessing professional healthcare support in suicidal crises, conflict with staff, and confidentiality, which restricted carer access to information that may have assisted with understanding suicidal behaviour [17]. This emerging evidence base provides some indication of the challenges faced by carers when supporting someone with psychosis or bipolar disorder and co-occurring suicidal behaviour. However, explanations for why carers are more likely to experience distress in the presence of suicidal behaviour amongst those they support, compared with carers of people with psychosis or bipolar disorder who do not experience suicidal behaviour [13,14,15], are limited. Further qualitative research focused on developing in-depth accounts of carers’ lived experiences in this context may assist with generating insights absent in the existing literature.

The aim of the current study was to understand the experience of caring for a family member with psychosis or bipolar disorder who has also experienced suicidal behaviour. In order to realise this aim, we conducted an exploratory qualitative investigation of data from the Relatives Education and Coping Toolkit (REACT) online forum. REACT is an online, peer-supported self-management intervention which aims to offer National Institute for Health and Care Excellence-recommended [10] education and emotional support to carers supporting people with psychosis or bipolar disorder. It was developed with extensive involvement from carers [7], has been evaluated in a large randomised controlled trial [18], and was used to identify critical factors impacting on the implementation of digital health intervention into UK mental health services [19]. REACT included a moderated online forum designed to facilitate peer support.

Online support forums are increasingly used by researchers to access naturalistic interactions focused on specific healthcare topics [20]. Factors such as the ability to post anonymously and the tendency for comments to be guided by forum users’ own priorities facilitate forms of personal disclosure less likely to occur in other contexts, such as research interviews [21]. Research with online mental health forums has highlighted how a forum’s culture and modes of interaction generate both peer support and interpersonal challenges [22,23], including for those experiencing suicidal behaviour [24,25], psychosis [26,27], and bipolar disorder [28,29]. Online forums for carers of people with mental health problems including psychosis have been developed and evaluated [30,31]. However, to date, they have not been used to inform qualitative research into suicidal behaviour and bipolar disorder or psychosis from carers’ perspectives. A recent quantitative, computational linguistic analysis of the entire REACT forum revealed “death and suicide” as one of five prominent thematic domains [32]. Other domains of discussion included negative emotions, conflict and abuse, illness and hospitalisation, and time. As such, this dataset represents a valuable information source for investigating carers’ experiences of supporting a family member with psychosis or bipolar disorder and suicidal behaviour.

## 2. Materials and Methods

### 2.1. Design

This qualitative study is a secondary analysis of forum posts collected as part of a randomised controlled trial to evaluate the REACT intervention [18]. Data were generated by forum users between April 2016 and June 2018.

### 2.2. Participants

To be eligible for the REACT trial, participants had to be aged 16 or over and live in the UK [18]. All participants self-identified as having a close friend or family member with psychosis or bipolar disorder. These broad inclusion criteria were justified on the basis that UK clinical guidance recommends dedicated psychoeducation and support for carers of people with any form of psychosis (including schizophrenia and related disorders) or bipolar disorder [10,33]. The research team recognise that these mental health experiences are not mutually exclusive; however, only the primary diagnoses of those being cared for were collected from participants in the REACT trial. All participants self-identified as help seeking and experienced high levels of distress associated with their friend or family member’s mental health, as indicated by a score of ≥3 on the General Health Questionnaire item “have you recently been feeling nervous and strung up all the time?” [34]. Access to the forum was restricted to participants in the intervention arm of the REACT trial (*n* = 399). All forum users were identified by a self-selected username that did not contain any personally identifiable information. Participants could contribute to the forum by writing messages, or “posts,” within forum “threads”, which were conversations visible to all forum users. The forum was moderated during working hours by REACT Supporters, who were family members or friends of people who had experienced psychosis or bipolar disorder. REACT Supporters were trained to provide emotional support and were supervised by a clinical supervisor (SJ) and the trial chief investigator (FL), both of whom are professors of clinical psychology. Participants also had access to a direct messaging function through which they could communicate with REACT Supporters. Direct messages were not visible to other forum users. Forum posts and direct messages were text-only and unrestricted in length.

### 2.3. Data Extraction

As the forum did not specifically direct participants to discuss suicidal behaviour, the following data extraction process was applied to identify conversations relevant to the research aim. As per a recent systematic review of caregiving experiences and suicidal behaviour, we applied a definition of suicidal behaviour that included any reference to suicidal thoughts, feelings/urges, plans, and/or attempts, in addition to self-injurious behaviour regardless of intent [12]. While it is acknowledged that approaches to the definition of suicidal behaviour and self-injury vary, the exploratory nature of this qualitative study justified a broad approach to data inclusion. To be eligible for inclusion in the analysis, forum posts related to suicidal behaviour were required to also refer to the experience of caregiving. Eligible posts could refer to present or past experience of suicidal behaviour. Abstract references to suicidal behaviour, or posts that only referred to carers’ own suicidal behaviour and not those of the person they supported, were excluded. To identify relevant data, the full REACT forum including all threads and direct message conversations was downloaded in Microsoft Word format. Each forum user was allocated a random participant (P) ID number. Multiple readings of the dataset were conducted by the first author (PM) to identify eligible forum posts. A second researcher (HR) then independently reviewed these candidate forum threads/direct message conversations to check their relevance to caregiving and suicidal behaviour. Differences were reviewed and resolved in subsequent conversation between researchers. Within each individual conversation, only forum posts written by participants who had referred to suicidal behaviour were coded as part of the analysis. Comments made by REACT Supporters were excluded.

### 2.4. Analysis

Data were analysed using thematic analysis from a critical realist perspective [35,36]. Critical realism combines ontological realism with epistemological relativism and takes the view that while a mind-independent social reality exists, analysis of this reality is necessarily mediated by the researcher’s idiosyncratic interpretive perspective. We therefore chose to apply the form of thematic analysis described by Braun and Clarke as “reflexive” thematic analysis [37], as part of which researcher interpretation is framed as an analytic resource rather than a source of undesirable bias. The first author is a PhD candidate in health research (PM), supported by an expert in the psychological science of suicide who is also an expert by experience (TG), academics with extensive research experience related to psychosis and bipolar disorder (FL, SJ, and HR), and clinical experience working professionally with service users with psychosis, bipolar disorder, and their family members (FL and SJ). Our aim was to use this diversity of experience and perspective to generate a nuanced account of participants’ experiences through iterative feedback on the developing analysis.

The analytic procedure followed guidance for reflexive thematic analysis [35,38]. First, PM conducted data familiarisation through multiple readings of extracted forum conversations, during which initial impressions and features of the data were noted. PM conducted initial coding by attaching brief labels capturing expressions of meaning to sections of forum text. Codes were generated inductively, that is, without reference to a pre-existing framework or theoretical constructs. PM generated initial sub-themes by identifying salient patterns of meaning across the dataset through an iterative process of reviewing underlying data, initial codes, and groups of codes that could be encompassed by overarching candidate themes. These candidate themes were refined based on feedback from the wider research team. A revised thematic structure was “sense-checked” via written feedback [38] by a REACT Supporter who was active on the forum throughout the delivery of the intervention. Analysis was conducted on NVivo 12 [39].

## 3. Results

The final dataset comprised 178 posts written by 29 forum users (Table 1). Posts appeared within 28 open forum threads and 8 direct message conversations. A majority of forum users were female (*n* = 26) and from a white British background (*n* = 25). The mean age of forum users was 49 years (range: 23–68). As indicated by Table 1, most (*n* = 24) forum users posted fewer than 10 times in reference to suicidal behaviour.

The analysis generated 3 themes, each with 2 sub-themes (Table 2).

### 3.1. Theme 1: Suicide as the “Ultimate Threat”

Carers’ forum posts highlighted their profound anxiety regarding the possibility that their family members may experience mental health crises involving suicidal behaviour. For one carer, this ongoing “threat” was informed by prior experience of suicidal crises: “*his [sic] a history of attempting suicide and we are so scared that we will lose him and that is his ultimate threat”* (P17).

#### 3.1.1. Sub-Theme 1.1: Living in Fear of Suicide

Carers’ fears about suicide were persistent and difficult to control. Distress appeared most prominent where carers felt that they were hopeless in the face of their family member’s deteriorating mental health: “*She won’t speak to anyone else except me…everything I do/say is wrong. She is again now telling me she wants to kill herself. I am so emotionally exhausted I don’t know what to do*” (P23). Such comments reflect a sense of desperation and fatigue which was particularly evident in carers who were providing support to a family member in crisis at the time of their posting to the forum.

A key factor that appeared to exacerbate this fear was carers’ difficulties with understanding what had caused the apparent escalation in the severity of their family member’s mental health difficulties. For example, one carer recalled that the difficulty of identifying a reason for their family member’s suicide attempt “*in a way scares me, as I think what was it all about*?” (P11). In the absence of an explanation for these experiences, carers expressed limitations in their perceived ability to manage suicidal behaviour in the community: “*When he was released from hospital, we were given no help or support from anybody, we were left living in terror, not knowing why this had occurred*” (P12). This immediate, terrified response to hospital discharge highlights how supporting a family member following the transition out of health services represents a potentially highly stressful period of uncertainty.

#### 3.1.2. Sub-Theme 1.2: Negotiating Responsibility for Living

The issue of responsibility for the welfare and safety of a family member was reflected in posts by forum users who struggled to establish a balance between their own wellbeing and provision of care. The emotional impact of feeling responsible for the life of a family member may be intensified in the context of parenthood. The socially salient expectation that parents should seek to provide life-maintaining support informed a uniquely isolating experience that detached one carer from their broader social network:
*“I know that there aren’t really many things other people can do to help with the pain and anguish, the worry of losing your children through suicide and the alienation you feel from your friends whose children are doing as you expected yours would do”*(P13)

In a similar vein, prioritising self-care may at times be necessary, but particularly difficult for parents where a “threat” of suicide exists. Negotiating a balance between these competing priorities was particularly difficult for one carer:
*“With the threat of self-harm or suicide as an action from them it’s always so hard to protect ourselves but also support and fight for them. What I am learning is we do have to be kind to ourselves and at times step back as a parent that is so so hard to do!”*(P19)

The type of relationship shared by carers and service users may frame the negotiation of caregiving responsibilities in the presence of suicidal behaviour. As one carer recalled regarding their partner: “*He was verbally abusive to me and I left. Not the first time this has happened. I came back because I was worried about his safety—he has attempted suicide twice in the past year*” (P28). In response, a carer offered support by drawing from their own lived experience:
*“My husband and I separated for a time when he was at his most ill, he made several suicide attempts during this time and I realised that this was not my fault, it was his illness. One of my conditions for getting back together was that he engaged with mental health services and another was that he took responsibility for his own mental health”*(P27)

This comment illustrates how recognising that the carer was not to blame for the situation, and that the service user had to some extent take responsibility for their own mental health, allowed this carer to re-negotiate the parameters of their personal relationship to protect their own wellbeing.

### 3.2. Theme 2: ‘Bouncing from One Crisis to Another’

Many carers had experienced multiple periods of intense distress. Participants’ average time spent providing care was over 9 years, during which many had lived through recurrent mental health crises within the family. Health services were largely viewed as insufficiently considerate of the role of the carer, both at the point of first contact during an emergency, and later, following transition to community care.

#### 3.2.1. Sub Theme 2.1: Responding to Crises

Carers recounted multiple experiences of supporting their family members through mental health crises: “*I’ve lost count of the number of suicide attempts and contacts with crisis team*” (P27). Rather than discrete events, crises were viewed as cyclical disruptions to normality and family functioning: “*I really know that feeling of ‘as soon as I start to relax something kicks off again’ as we lived with that for so long*” (P27). Suicidal crises were characterised by a heightened sense of urgency, where immediate risk of harm demanded carers take action to ensure their family member’s safety. One carer recalled directly intervening to prevent their sibling from engaging in self-injury, which left a lasting impact on the family: “*I was hanging on to him to try and prevent him harming himself. It was a terrifying experience from which we have never recovered”* (P12). Carers’ initial actions during crises often included contacting emergency services, primarily the police. Some appreciated that the police were reliable sources of support in dangerous situations and could facilitate access to medical treatment:
*“Another avenue I have found helpful is 111 [non-emergency telephone health service] to get and [sic] out of hours GP to come and assess my husband at home when things were really bad one evening...It was the police that suggested that. To be honest most of what I have learned about negotiating mental health crisis services has been from the police”*(P27)

Carers’ own primary motivations overlapped with those of the police, namely, establishing the immediate physical safety of those involved. However, seeking help in this way created additional challenges that could contribute to distressing encounters:
*“It was extremely traumatic as they [police] came ‘mob handed’ expecting a fight but were eventually persuaded to let me take her to the hospital in my car, which was probably good for her but didn’t do a lot for my feelings of guilt”*(P18)

Carers expressed mixed views on support offered by crisis and emergency health services. One forum user saw clear value in seeking admission and anti-psychotic medication in crisis situations, “*He went into hospital because voices were telling him to kill himself which he tried to do, and these have been stopped with meds*” (P22). However, this could be followed by challenges with medication adherence post-discharge:
*“He was completely psychotic, with no insight and had just been discharged from hospital following a suicide attempt, yet there seemed to be no concern that he hadn’t been to collect his meds or had any contact with crisis team”*(P27)

#### 3.2.2. Sub-Theme: 2.2 Being Left ‘at a Loss’ about What to Do Next

Carers expressed how post-crisis transitions to community-based support had been rushed or unsupported, an experience which “*leaves families, who are often terrified and at a loss themselves, clueless how to help*” (P12). Transition points in and out of services were particularly problematic. Carers were concerned that those who were voluntarily admitted to hospital could leave of their own volition:
*“by the time they [A&E staff] actually see him he will generally say ‘No I don’t want to kill myself I was just being stupid’ (even when it’s been the second time that day) and has just been referred back to his GP with probable depression”*(P27)

Community-based support was valued where available but seen by some as limited in scope due to the challenge of maintaining contact with health services over time, *“He had about four meet ups with his care worker but no care plan that I was aware of nor any constructive support”* (P13). This absence of ongoing health service contact served to reset the cyclical recurrence of escalating mental health difficulties, in which carers are left without support:
*“The crisis team and local crisis centre have been good when he’s come under their wing a few times, but he has been very quickly discharged from their care once the crisis is over and that’s where we lose contact”*(P20)

### 3.3. Theme 3 “It Definitely Needs to Be Easier to Get Help”

This final theme highlights the ways in which forum users struggled to access mental health support, underpinned by factors such as health service emphasis on confidentiality at the expense of carer involvement and the suggestion that a mental health crisis is necessary to initiate healthcare support: “*the thing that makes me most angry is that my husband had to reach crisis point before getting any help*” (P27). Participants appeared to use the peer-support forum to fill gaps in their own self-care and help-seeking strategies.

#### 3.3.1. Sub-Theme 3.1: Being (Un)Involved in Professional Care

Whilst seeking help, carers acted as mediators positioned between professionals and their family members, aiming to promote interactions and engagement between the two. Yet where contact with health services had been made, some found healthcare professionals distant and their decision-making processes unclear. Carers had intimate knowledge of their family members’ personal histories and, often, years of experience supporting them. This expertise was not always acknowledged and used by professionals, resulting in some carers feeling that their potential contribution to their family member’s care had been overlooked. Discord between professional and carer perspectives was pronounced where carers’ concerns about suicidal behaviour were not reflected in professionals’ decisions:
“*She tried to take an overdose 4 weeks ago. They are now saying that she can leave tomorrow. Her husband is very concerned, as to how he treats the situation. Does he just let her get on with her ‘life’ or has he always got to be there watching what she is doing*?”(P11)

In contrast, being actively involved in the care process is highly valued by carers: “*I’m so lucky with my husband’s CPN [Community Psychiatric Nurse] as he says that keeping communication open with me is in my husband’s best interest and that as he has stated when he is well that he wants me involved he is following his wishes when he lacks capacity to make the decision*.” (P27)

This involvement was particularly appreciated by the same carer, who explained how psychotic experiences can interact with suicidal behaviour to produce additional challenges with professional help-seeking:
*“he thought…that I was an undercover police officer monitoring him, that I was trying to poison him… on one occasion that week things were getting really bad [sic] get him in to A&E to speak to crisis team but he would only tell them about feeling suicidal and sent me out when I tried to explain what was really happening.”*(P27)

Health service confidentiality and the requirement for service users to pro-actively engage with health services represented barriers for carers seeking support. Strict adherence to these principles was perceived by two carers as incongruent with the severity of the situation:
“*he has recently felt suicidal. When I called the hospital for help they told me that he would have to ring himself and they couldn’t help*” (P26) and “*had a bad weekend. Daughter would not consent to crisis team on Saturday. She had distressing voices telling her to kill herself. On and On. I was very anxious after she told me this.*”(P3)

This placed some carers in a paradoxical situation, where a lack of motivation to engage with services seemed to be attributable to the very mental health difficulties carers sought help for:
“*They said ‘yes he sounds very poorly to us but unless he wants us involved, we can’t do anything’. OK so he was completely delusional and had no insight into the fact he was ill, at what point were they expecting him to say: ‘oh yes please I’d like some help with my mental health?’*”(P27)

#### 3.3.2. Sub-Theme 3.2: Peers Address Unmet Support Needs

Carers’ reflected on the importance of having a space to share their experiences and access validating accounts of others’ similar experiences, opportunities that were evidently not often present within some carers’ wider lives: “*I tend to write about all the bad bits here because it’s the only place I can*” (P27). For one forum user, the presence of peers whose life experiences resonated with their own provided *“relief that I am no longer feeling so alone and isolated dealing with my son*” (P20). Particularly welcome was the use of personal experience to inform advice, in comments such as *“my advice to anyone supporting another person is to put your own mental and physical health first. I have learnt this the hard way*” (P27). Indeed, those who had been through similar difficult situations readily offered valued insight beyond what appeared to be available outside of the forum.

## 4. Discussion

The aim of this study was to understand carers’ experiences of supporting people with psychosis or bipolar disorder who have also experienced suicidal behaviour. To our knowledge, this is the first use of comments made on an online peer-support forum to investigate these specific caregiving experiences. There were several important findings across three overarching themes. First, forum posts revealed how carers experienced a sense of personal responsibility for the lives of those receiving support, alongside a difficulty with understanding and managing suicidal behaviour. This engendered significant stress and ongoing fear across carers’ lives. A second theme highlighted how many had supported their family members through multiple crises and had largely found professional support for carers to be unsatisfactory, especially at points of transition between health services and the community. Some carers saw their attempts to seek help rebuffed by health professionals due to their inflexible application of confidentiality procedures or lack of consent and engagement amongst their family members. A third theme highlighted how carers’ often unmet desires for ready access to collaborative health services in suicidal situations. Within this context, carers found relief and comfort in an online community of peers, which represented a safe place for sharing and accessing lived experiences that resonated with their own.

Previous quantitative analysis of the REACT forum highlighted how carers used the service to connect with peers over challenging aspects of their lives, including ongoing stressful events, conflict, and suicide [32]. Our focused analysis of posts related to suicidal behaviour of those receiving support highlighted how, consistent with previous research, carers live with intense fear regarding the possibility that their family members may experience further suicidal behaviour [40,41,42]. Distress was particularly prominent during situations in which carers found it difficult to understand the reason for, and how to reduce, the likelihood of suicidal behaviour. Findings from this study add context to quantitative evidence showing elevated carer distress amongst families of people with psychosis who have also experienced suicidal behaviour, relative to those without prior experience of suicidal behaviour [13,43]. This literature suggests that suicidal behaviour in psychosis is associated with poorer carer quality of life across all life domains, lower family functioning, and more negative appraisals of caregiving [14,43,44]. Evidence presented here suggests that this broad psychosocial impact may be related to the uniquely pervasive anxiety apparent in the lives of carers of people experiencing suicidal behaviour (theme 1), repeated suicidal crises and limited support between them (theme 2), and lack of access to carer-inclusive professional support (theme 3). Prior research also indicates that beliefs in greater future severity and lower controllability of psychosis are linked to greater carer distress [45]. One interpretation of carers’ lived experiences described here is that ongoing fear of further suicidal behaviour and challenges with understanding how to mitigate its reoccurrence exacerbate these appraisals and thus contribute to more severe impacts on carer wellbeing. These psychological processes represent potential but as yet under-investigated mechanisms by which carers may experience poorer psychosocial outcomes in the presence of suicidal behaviour.

As in a prior study with families of people with bipolar disorder receiving healthcare support following suicidal behaviour [17], health services were described as unable or unwilling to include carers in key decisions regarding their family member’s care or provide information regarding how carers could effectively support their family members. Findings of the current study also indicate that feeling unsupported by health services when caring for someone experiencing suicidal behaviour exacerbates carer distress, especially in circumstances where carers feel that their potential involvement in care is overlooked. Individualised care plans that are developed collaboratively alongside carers and draw on their knowledge of their family member’s circumstances represent one strategy for addressing carers’ desires for greater involvement in healthcare delivery [46]. However, a systematic review of families’ experiences of involvement in mental health care planning highlighted a number of barriers consistent with carers experiences in the current study [47]. This included health professionals’ underappreciation of families’ lived experiences and approaches to information sharing that limited valid and mutually beneficial interactions between carers and healthcare staff. Contemporary policies (in the UK) do, however, allow for constructive information sharing between professionals and carers that highlights carers’ needs within the boundaries of service users’ rights [48]. Best practices for suicide prevention recommend establishing information-sharing principles with families early in the care process, including identifying circumstances under which information would be provided to families if service users lose capacity to consent to information being shared [49]. Future research may seek to evaluate barriers to the implementation of initiatives that may address these challenges.

The absence of service user consent to engage voluntarily with healthcare support can present carers of people experiencing suicidal behaviour with a significant obstacle to help-seeking [50]. In this study, two participants noted how their family member’s experience of psychosis exacerbated this challenge. Attempting to understand delusional beliefs, especially those focused on carers, can be a particularly difficult and distressing task that involves attempts to piece together a coherent understanding of others’ mental states [51]. However, carers are likely to have extensive experience of, and insight into, the nature of their family member’s mental health experiences, which may be invaluable to the provision of appropriate mental health care. Including carers in professional support would likely rely on open communication with mental health professionals, yet as has been reported in previous research [52], some carers in the current study noted that service user confidentiality acted as a barrier to accessing information about their care. Guidance related to this issue suggests that clinical judgements about information sharing should prioritise harm-reduction, and where information sharing is not appropriate, clinicians should engage with carers regarding their own support needs [53]. This could involve referral for a dedicated carer’s assessment [54]. Additional qualitative work drawing on multi-stakeholder perspectives may be of value in identifying how these best practices can be effectively applied in suicidal crises, in a way that accounts for the views of service users, the needs of carers, and the professional obligations of healthcare staff.

Carers’ reflections on the cyclical and demanding nature of crisis situations align with previous research indicating that these periods are especially challenging [55]. Mental health crises, and particularly those in which there is concern about suicide, are characterised by conflicting emotions in carers regarding how to respond, apprehension concerning police intervention, and the experience of being “invisible experts” regarding their family member’s mental health [55]. Indeed, in the present study, forum users noted a tension between the need to engage with emergency services and the potential for distress during a family member’s involuntary admission to health services. Efforts have been made to embed mental health expertise within police responses to mental health crises, including as part of “co-response” teams comprising mental health professionals and law enforcement personnel. However, a recent systematic review [56] of related literature reported that while co-responder models show improvement in some crisis outcomes relative to police-only models, such as arrest rate, the current evidence base provides mixed support for their overall effectiveness and is limited to largely low-quality studies. The extent to which these models contribute to improved outcomes for families of people experiencing crisis is currently unclear. This is significant given evidence of service users’ preferences for the involvement of family members rather than police personnel in crisis situations [57] and carers’ reflections on the frightening and sometimes traumatising nature of mental health crises involving police intervention [58,59]. Evidence presented here supports the recommendation [56] for research into co-designed crisis models and subsequent large-scale evaluation to address this highly significant point in the care pathway for people experiencing mental health crises and their families.

As described here, many carers supporting people with mental health difficulties experience barriers to carer-focused information and support. A scoping review of international research [60] revealed barriers at the level of the individual carer (low expectations regarding involvement in care and unequal power relationships with health professionals) and healthcare organisations (lack of carer-focused service provision and concern over the impact of family involvement). The World Health Organisation’s comprehensive mental health action plan (2013–2030) calls for the strengthening of carer involvement in the design and delivery of integrated healthcare systems [61]. The results of this study highlight implications for service development at multiple levels of provision. Digital interventions are feasible and hold potential for the efficient delivery of rigorously developed psychoeducation and peer support [62]. Yet, as was highlighted by a multiple case study of REACT implementation, such services require extensive and iterative support to promote their integration within and uptake by health services [19]. Carer-inclusive information-sharing and support is also desirable at the point of contact with health services. Indeed, health professionals working in suicide prevention have identified that developing stronger links with families would enhance their ability to deliver safe care for those experiencing suicidal behaviour [63]. Brief evidence-based strategies for carer involvement in this context include safety planning, which could be used to highlight the role of carers in suicide prevention and important factors such as managing lethal means at home [64,65]. At a systemic level, alternative models of mental health care that foreground social relationships are likely to promote carer engagement. One example is Open Dialogue, which frames the service user and their social network as the focus of intervention [66]. Crisis care therefore involves extensive carer involvement, inclusive information sharing, and collaborative decision making. Increased implementation of these principles is likely to address carers’ feelings of exclusion from the care process and reported difficulties with accessing information about the care their family members receive.

### Strengths and Limitations

A moderated online forum designed to elicit peer support amongst carers of people with psychosis and bipolar disorder represents a novel source of data for understanding carers’ experiences and support needs, independent from the somewhat artificially constructed context of other research settings. Posts focusing on suicide related content were extracted, offering an opportunity to explore an important and under-researched issue. Analysis was informed by a range of professional and lived expertise. However, there were some limitations. While an inclusive approach to the identification of relevant forum conversations was applied, including terms such as “self-harm”, it is appropriate to acknowledge that not all self-injury is motivated by an intent to cause death [67]. The sample in this study was UK-based, predominantly female, White British, and IT literate. As such, carers’ experiences may not align with those of other groups, such as ethnic minorities or those without access to online resources. A further limitation is that a large number of forum posts were written by a minority of users. This reflects the tendency in online forums, including those designed for mental health support, for a small number of “superusers” to generate the vast majority of forum posts [68]. Indeed, on the REACT forum as a whole, 93% of forum users posted five times or fewer [32].

## 5. Conclusions

This study is the first to draw on online forum data to investigate carers’ perspectives of supporting people with psychosis and bipolar disorder who have also experienced suicidal behaviour. Findings indicate that greater attention should be paid to understanding how carers can be assisted at each point in the care pathway, from the initial emergence of suicidal behaviour through to post-crisis care. A key challenge for both researchers and practitioners lies in designing strategies that can meet these goals and which account for the needs and expertise of carers, whilst also respecting the rights and wishes of service users within healthcare settings.

## Figures and Tables

**Table 1 ijerph-19-15192-t001:** Forum user demographic information.

ID	Age	Gender	Ethnicity	Relationship	Primary Diagnosis of Person Being Cared for	Time Caring (Years, Months)	Forum Posts Analysed
P1	54	Female	White British	Not given	Not given	24, 7	1
P2	50	Male	White British	Partner	Bipolar disorder	4, 0	10
P3	65	Female	White British	Mother	Bipolar disorder	20, 0	21
P4	25	Female	White British	Daughter	Bipolar disorder	18, 7	2
P5	45	Female	White British	Mother	Bipolar disorder	14, 0	1
P6	23	Female	Mixed	Daughter	Bipolar disorder	7, 0	1
P7	44	Female	White British	Partner	Bipolar disorder	3, 0	1
P8	47	Female	White British	Mother	Psychosis	3, 11	1
P9	65	Female	Irish	Mother	Schizophrenia	15, 6	1
P10	63	Female	White British	Mother	Bipolar disorder	24, 11	4
P11	57	Female	White British	Sibling	Psychosis	0, 4	4
P12	50	Female	White British	Sibling	Schizoaffective disorder	20, 0	2
P13	60	Female	White British	Mother	Bipolar disorder	12, 0	7
P14	58	Male	White British	Partner	Bipolar disorder	15, 0	2
P15	68	Female	White: Other	Not given	Not given	19, 0	1
P16	45	Female	White British	Partner	Bipolar disorder	17, 10	19
P17	60	Female	White British	Mother	Bipolar disorder	11, 9	1
P18	63	Male	White British	Partner	Psychosis	3, 0	1
P19	52	Female	White British	Mother	Bipolar disorder	3, 0	5
P20	51	Female	White British	Not given	Not given	11, 0	1
P21	30	Female	White British	Daughter	Bipolar disorder	9, 3	1
P22	45	Female	White British	Partner	Bipolar disorder	1, 6	5
P23	51	Female	White British	Partner	Psychosis	3, 0	5
P24	37	Female	White British	Partner	Schizoaffective disorder	16, 0	13
P25	60	Female	White: Other	Mother	Schizophrenia	6, 9	1
P26	42	Female	White British	Partner	Bipolar disorder	5, 8	2
P27	34	Female	White British	Not given	Not given	9, 4	58
P28	33	Female	White British	Partner	Schizoaffective disorder	7, 6	6
P29	50	Female	White British	Partner	Psychosis	0, 6	1

**Table 2 ijerph-19-15192-t002:** Table of themes and sub-themes.

Themes	Sub-Themes	Illustrative Quote
1. Suicide as the ‘ultimate threat’	1.1. Living in fear of suicide	*“I worry constantly about his mental well-being and safety as he has talked about suicide on several occasions in the past. I feel like I’m always waiting for ‘that phone call’”* (P20).
	1.2. Negotiating responsibility for living	“*Things cannot get much worse and maybe the kinder option would have been to not interrupt his plan to die, at least he would be at peace but then I doubt I could live with that*” (P10).
2. ‘Bouncing from one crisis to another’	2.1. Responding to crises	*“I have spent several nights sitting in A&E [accident and emergency] with my daughter in an agitated/manic state…I feel the environment only contributed to her state and the wait and the busy environment only increased my own state of anxiety which was high already. I have to comment that we have experienced several different hospitals and the situation is similar in all…”* (P19).
	2.2. Being left ‘at a loss’ about what to do next	*“…he walked out [of hospital] yesterday with no care plan and no idea of support”* (P13).
3. ‘It definitely needs to be easier to get help’	3.1. Being (un)involved in professional care	“*I really feel like no one wants to speak to me about my husband and it makes me feel very guilty as if I’m making it up*” (P22).
	3.2. Peers address unmet support needs	“*I have literally just joined and already feel such relief that I have somewhere to ask these questions and find information*” (P8).

## Data Availability

Ownership of copyright and intellectual property rights for all research conducted for the REACT trial will ultimately be held by Lancaster University. We intend to make available the individual participant data that underlie the results reported in this article, after de-identification. Data will be made available on request 12 months following article publication, and only to researchers who provide a methodologically sound proposal and where the proposed use of the data has been approved by an independent ethics review committee (‘learned intermediary’) identified for this purpose. Proposals should be directed to the corresponding author at rdm@lancaster.ac.uk. Data will be available for 10 years at Lancaster University’s Research Directory (https://doi.org/10.17635/lancaster/researchdata/306, accessed on 17 April 2019). The study protocol and statistical analysis plan are already published and freely available.

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
