# Peer review of "Caring for a Family Member with Psychosis or Bipolar Disorder Who Has Experienced Suicidal Behaviour: An Exploratory Qualitative Study of an Online Peer-Support Forum"

_ijerph, 2022, doi:10.3390/ijerph192215192_

Round 1

Reviewer 1 Report

Thank you for you original research. Method section is very nice, appropriate an d clear.
Results section ok but not al quotes are in italic. 
In the discussion section I think you can elaborate more on the dramatic issue that the barrier to get professional help is so high, even with the consequence that 'the police is a worthwhile source of information'! Why is , all over Europe, this issue so common? Can you comment on that and therefore also in the abstract under 'conclusion' mention this issue. The system itself causes problems, the element 'privacy' and gdpr rules are used in my opinion (I am a GP) a wrong way. Are alternatives way of handling possible?  And what is the evidence? 

Author Response

Thank you for your review and the opportunity to improve this article. I have addressed your points individually below:

Point 1: Thank you for you original research. Method section is very nice, appropriate an d clear. Results section ok but not al quotes are in italic. 

Response: All quotes have been italicised

Point 2: In the discussion section I think you can elaborate more on the dramatic issue that the barrier to get professional help is so high, even with the consequence that 'the police is a worthwhile source of information'! Why is , all over Europe, this issue so common? Can you comment on that and therefore also in the abstract under 'conclusion' mention this issue. The system itself causes problems, the element 'privacy' and gdpr rules are used in my opinion (I am a GP) a wrong way. Are alternatives way of handling possible?  And what is the evidence? 

Response: I have added a paragraph to the discussion that acknowledges the common and international challenges carers face when attempting to access information and support. I have commented specifically on factors that may address this issue including uptake of digital interventions, brief planning interventions focused on suicide prevention, and an alternative approach to of mental health care with greater emphasis on the role of carers (Open Dialogue). I have highlighted these approaches in the conclusions section of the abstract. I agree that issues related to privacy and information sharing are relevant here. I have expanded on this in the third paragraph of the discussion.

Reviewer 2 Report

The manuscript faces an interesting topic for psychiatrist as caregivers of suicidal patients are often an understimate resource for this high risk patients.

I've only two comments for authors:

1) patients are divided in two main groups: bipolar disorders and Psychosis. The latter sounds to me too aspecific: what authors mean with the term psychosis (also bipolar disorder I is a psychotic disorder)

2) the results section seems to me too long and narrative. I think it could be shortened with use of tables or with more datat aggregation

Author Response

Thank you for your review and the opportunity to improve this article. I have addressed your comments as below:

Point 1: patients are divided in two main groups: bipolar disorders and Psychosis. The latter sounds to me too aspecific: what authors mean with the term psychosis (also bipolar disorder I is a psychotic disorder)

I agree that the category ‘psychosis’ is broad. I have added a sentence noting that broad inclusion criteria were justified on the basis that UK clinical guidance recommends psychoeducation and emotional support for carers of people with any form of psychosis (including schizophrenia and related disorders) and bipolar disorder. I have also clarified that carers eligible for the REACT study self-identified as having supported someone with psychosis or bipolar disorder (specific primary diagnoses are presented in table 1). I have noted that the research team recognise that psychosis and bipolar experiences are not mutually exclusive, however, only primary diagnoses were collected as part of the REACT trial.

Point 2: the results section seems to me too long and narrative. I think it could be shortened with use of tables or with more datat aggregation

I have moved several quotes from the main body of the results section to table 2. They are included in the table alongside sub-theme titles as illustrative quotes. This has reduced the overall length of the main body of the results section by approximately 300 words.

Round 2

Reviewer 1 Report

Thank you for considering the remarks and add a sustantia answer in the discussion section